# Flavonoid-Modifying Capabilities of the Human Gut Microbiome—An In Silico Study

**DOI:** 10.3390/nu13082688

**Published:** 2021-08-03

**Authors:** Tobias Goris, Rafael R. C. Cuadrat, Annett Braune

**Affiliations:** 1Research Group Intestinal Microbiology, Department of Molecular Toxicology, German Institute of Human Nutrition Potsdam-Rehbruecke, Arthur-Scheunert-Allee 114-116, 14558 Nuthetal, Germany; braune@dife.de; 2Department of Molecular Epidemiology, German Institute of Human Nutrition Potsdam-Rehbruecke, Arthur-Scheunert-Allee 114-116, 14558 Nuthetal, Germany; Rafael.Cuadrat@dife.de

**Keywords:** polyphenols, plant metabolites, phytohormones, isoflavones, personalized nutrition, metagenomes, human gut microbiome

## Abstract

Flavonoids are a major group of dietary plant polyphenols and have a positive health impact, but their modification and degradation in the human gut is still widely unknown. Due to the rise of metagenome data of the human gut microbiome and the assembly of hundreds of thousands of bacterial metagenome-assembled genomes (MAGs), large-scale screening for potential flavonoid-modifying enzymes of human gut bacteria is now feasible. With sequences of characterized flavonoid-transforming enzymes as queries, the Unified Human Gastrointestinal Protein catalog was analyzed and genes encoding putative flavonoid-modifying enzymes were quantified. The results revealed that flavonoid-modifying enzymes are often encoded in gut bacteria hitherto not considered to modify flavonoids. The enzymes for the physiologically important daidzein-to-equol conversion, well studied in *Slackia*
*isoflavoniconvertens*, were encoded only to a minor extent in *Slackia* MAGs, but were more abundant in *Adlercreutzia equolifaciens* and an uncharacterized *Eggerthellaceae* species. In addition, enzymes with a sequence identity of about 35% were encoded in highly abundant MAGs of uncultivated *Collinsella* species, which suggests a hitherto uncharacterized daidzein-to-equol potential in these bacteria. Of all potential flavonoid modification steps, *O*-deglycosylation (including derhamnosylation) was by far the most abundant in this analysis. In contrast, enzymes putatively involved in *C*-deglycosylation were detected less often in human gut bacteria and mainly found in *Agathobacter faecis* (formerly *Roseburia faecis*). Homologs to phloretin hydrolase, flavanonol/flavanone-cleaving reductase and flavone reductase were of intermediate abundance (several hundred MAGs) and mainly prevalent in *Flavonifractor plautii*. This first comprehensive insight into the black box of flavonoid modification in the human gut highlights many hitherto overlooked and uncultured bacterial genera and species as potential key organisms in flavonoid modification. This could lead to a significant contribution to future biochemical-microbiological investigations on gut bacterial flavonoid transformation. In addition, our results are important for individual nutritional recommendations and for biotechnological applications that rely on novel enzymes catalyzing potentially useful flavonoid modification reactions.

## 1. Introduction

Flavonoids, which are exclusively plant secondary metabolites, are considered to bear many beneficial effects on health [1,2,3,4,5] and are assumed to contribute especially to a lower cardiovascular disease and cancer-related mortality [6,7]. Most flavonoids are taken up as a part of the human diet mainly as glycosides. A portion of dietary flavonoids is absorbed in the small intestine following *O*-deglycosylation by epithelial enzymes. Flavonoid aglycons then undergo phase I/II transformation [8,9,10]. The major part of dietary flavonoids (or their phase I/II metabolites) reach the large intestine, where they are subject to transformation by gut bacteria which can enhance or lower their bioavailability and biological activtity [11,12,13,14]. In particular, members of the genera *Bifidobacterium* and *Lactobacillus* were observed to *O*-deglycosylate, e.g., flavanone and isoflavone glycosides [11]. While *O*-deglycosylation was the most extensively described flavonoid deglycosylation, with many glycosides from flavonols (e.g., rutin in several vegetables), flavanones (often found in citrus fruits) and isoflavones (present mainly in soy products) as substrates, several bacteria also mediate *C*-deglycosylation from compounds such as isovitexin, e.g., from buckwheat. Additional flavonoid transformations involve demethylation, dehydroxylation, reduction, and C-ring cleavage [11].

Many described gut bacterial flavonoid-converting activities lack biochemical or genetic analyses and, therefore, sequences of the involved enzymes. Of the *O*-deglycosylating enzymes, six beta-galactosidases [15,16,17] and a set of flavonoid-active rhamnosidases [18] that cleave the terminal rhamnose subunit of flavonoid rhamnosides (e.g., rutin or naringin), resulting in the corresponding flavonoid glucosides [19,20,21,22,23,24], have been characterized (Table 1). The more complex *C*-deglycosylating enzyme systems were studied in fewer gut bacteria [25,26,27] (Table 1). The *C*-deglycosylation system described in strain PUE is dependent on at least three enzymes, DgpABC, which are part of a large gene cluster including transporters and other accessory genes whose function in *C*-deglycosylation is unclear [26,27]. The *C*-deglycosylating enzyme system of *Eubacterium cellulosolvens* includes the five enzymes DfgABCDE [25]. Both corresponding enzyme complexes include at least one glycosyltransferase and an oxidoreductase. Phloretin hydrolysis as carried out by phloretin hydrolase [28,29] and the three-step conversion of daidzein to equol [30,31,32] are relatively well studied, while the gut bacterial enzymes involved in dehydroxylation and *O*-demethylation of flavonoids are not yet described. However, an *O*-demethylase from *Eubacterium limosum* ZL-II demethylates polyphenolic lignans, [33]. As with other characterized *O*-demethylases [34,35], this enzyme system consisted of two methyltransferases (MT), a corroinoid protein (CP) and an activating enzyme (AE), and might also be responsible for the *O*-demethylation of flavonoid carried out by *E. limosum* and *E. ramulus* [36,37].

The enzymes involved in daidzein-to-equol conversion and their encoding genes have been characterized in *Slackia eggerthella* and *Lactococcus* strains [30,32,38,39]. At least three enzymes are involved in the complete conversion of daidzein to equol. All three were proposed to be reductases [30,32], although one of them was recently suggested to be a dismutase [38]. In addition, a dihydrodaidzein (DHD) racemase is employed for stereochemical conversion of DHD in some bacteria [30]. Besides the complete conversion from daidzein to equol estimated to occur in about half of the population [40,41,42], a large number of individuals carry bacteria catalyzing only the reduction of daidzein to DHD or *O*-desmethylangolensin [42,43].

Phloretin hydrolase (Phy) hydrolytically cleaves the C-C bond adjacent to the aromatic A-ring of phloretin and thereby produces phloroglucinol and 3-(4-hydroxyphenyl)propionic acid. The first Phy was discovered in *E. ramulus* [28,44]. The only other Phy has been found in the intestinal pathogenic *Mycobacterium abscessis*, showing a 30% amino acid sequence identity to that of *E. ramulus* [27].

The flavanone- and flavanol-cleaving reductase (Fcr), cleaving the heterocyclic C-ring of flavanones and flavanonols, was recently characterized in detail [45]. Earlier suggested to be an enoate reductase and to act in a concerted pathway together with a chalcone isomerase (CHI, [46]), the Fcr of *E. ramulus* was described to catalyze the conversion of, e.g., naringenin to phloretin also without CHI. Still, as CHIs [46,47,48,49] are responsible for the production of auronol from flavanonols [47], they could play a substantial role in flavonoid transformation in the human gut. Very recently, the characterization of an ene-reductase (Flr), catalyzing the reduction of the C ring double bond of flavones and flavonols, was published [50].

All of the above studies were performed with few flavonoid-modifying bacteria and the corresponding enzymes originate from a limited number of bacterial species. Isolation of these strains from fecal samples could be biased because of selective enrichment of culturable bacteria. Therefore, the research on gut-bacterial flavonoid modification published up to now does not display a complete picture of flavonoid transformation in the human gut: phenotypic studies on human fecal samples rely mainly on (partial) 16S rRNA gene sequencing and do not resolve bacterial species. Thus, flavonoid-converting species might be missed. In addition, quantitative studies of flavonoid-converting bacteria are scarce. A recent study on the quantification of daidzein-to-equol-converting bacteria relied on primers targeting genes of involved reductases [52]. With the advent of metagenomic studies, a closer look into the gene catalog of the human gut microbiome became feasible [53], but still, quantification and especially taxonomic affiliation of the genes of interest are challenging. Current bioinformatic methods offered the opportunity to assemble genomes from metagenomic data [54], which were also recently applied to human gut metagenome studies [55,56,57], opening up new ways to reveal flavonoid modification in the human gut.

Here, we aim to identify and quantify flavonoid-modifying bacteria in the human gut microbiota. For this, we screened the most recent combined human gut MAGs collection, the Unified Human Gastrointestinal Protein or Genome (UHGP/UHGG) catalog of more than 280,000 assembled genomes, with a BLAST search using protein sequences of characterized flavonoid-modifying enzymes (Table 1) as queries. In addition, the presence, prevalence and abundance of bacterial species described to modify flavonoids was investigated. Based on the obtained data, a possible scenario of flavonoid modification by the identified human gut bacteria is presented.

## 2. Results and Discussion

### 2.1. Distribution of Flavonoid-Modifying Bacterial Species across Human Gut MAGs

First, we screened the accompanying metadata of the MAGs database for the abundance of bacterial species described to transform flavonoids. For this, a literature-based overview of currently known flavonoid-converting bacterial strains present in the human gut was compiled, resulting in 45 distinct bacterial strains of 43 different species (Table 2). Of these, eleven were not taxonomically validly published, either due to lack of a 16S rRNA gene sequence or because the authors chose an informal taxonomic designation. If a 16S rRNA sequence was available, these unclassified isolates were provisionally assigned to the closest taxonomically described species using a 98% sequence identity cutoff to the respective strain. *O*-deglycosylation was the most abundant flavonoid modification performed by human gut bacteria (31 strains). *C*-deglyocsylation, C-ring cleavage, reduction, dehydroxylation or *O*-demethylation of flavonoids were far less abundant (between three and seven strains each).

We then performed a text-based filtering and counting of species in the lineage column of the UHGG metadata with the aim of specifying the MAG abundances of described flavonoid-transforming species as percentages of the overall number of MAGs in the human gut. Important to note here is that the taxonomic assignments were based on the genome taxonomy database (GTDB, https://gtdb.ecogenomic.org/, accessed on 1 July 2020) in the UHGG metadata. Thus, species without a sequenced genome (*Catenibacillus scindens*, *Lactobacillus leichmanii* and *Slackia* sp. NATTS) or species lacking taxonomic assignment in the GTDB to a given genome (*S. isoflavoniconvertens*) could not be found. Besides the overall abundance of MAGs, we also calculated the prevalence of MAGs across individuals as percentages of the overall number of metagenome samples in the metadata.

The most prevalent flavonoid-modifying bacteria were *Bacteroides uniformis* (18.5% prevalence across all 21,508 individuals), *Parabacteroides distasonis* (14.5%), *Bifidobacterium infantis* (8.4%), and *Bi. adolescentis* (7.4%; Table 2). All of these species were described to be capable of flavonoid *O*-deglycosylation, mainly flavonols, isoflavones and flavanones [11]. The remaining species were detected in less than 5% of individuals (Table 2). The overall abundance across all MAGs were similarly distributed (Appendix A). The numbers for a single *E. coli* strain described to O-deglycosylate flavonoids [58] should be handled with care, since this activity was as-yet not confirmed for other *E. coli* strains and the MAGs database does not allow for strain-level assignment. The lack of *Bi. longum* in the MAGs metadata is remarkable, but can be explained by the fact that *Bi. longum* ssp. *infantis* [59] is classified as *Bi. infantis* in the GTDB. In addition, potential flavonoid-modifying bacterial species from a recently published BRENDA enzyme reaction database search [60] were screened for their abundance in the MAGs metadata. However, these bacteria showed only very low abundance (Appendix A) and were not included in this study.

### 2.2. Screening the MAGs Amino Acid Sequence Database for Potential Flavonoid-Modifying Enzymes

In the following, the UHGP database was screened with sequences from characterized flavonoid-modifying enzyme as queries. In total, 1,586,499 hits were found with an e-value of less than 1e-60, 1e-25 or 1e-20, filtered for a coverage of at least 75% and a PID of 30. In particular, *O*-glycosidases and rhamnosidases gave a large number of hits, while *O*-demethylases and *C*-glycosidases resulted in a rather low number of hits, because we filtered for the abundance of several genes from a larger gene cluster in the single genomes (Appendix A, pivot table).

### 2.3. O-Deglycosylation of Flavonoids

The five bifidobacterial flavonoid *O*-deglycosylating enzyme sequences, which showed a maximum amino acid percent sequence identity (PID) of 35 to each other, resulted in a high number of hits in MAGs classified to six different *Bifidobacterium* sp. (Figure 1) (PID threshold of 50, occurrence of at least 50 for each sequence hit). The highest numbers were observed in MAGs classified as *Bi. adolescentis* (overall, 6633 with a PID > 40), *Bi. infantis* and *Bi. pseudocatelunatum* (overall, about 3000 each, see also Appendix A). The *Ba. thetaiotaomicron O*-glycosidase query resulted mainly in MAGs from six different *Bacteroides* species, with *Ba. ovatus* and *Ba. thetaiotaomicron* (both about 500 occurrences) as the most numerous (Figure 1, Appendix A). *Bacteroides* members not already characterized as flavonoid *O*-deglycosylating species include *Ba. cellulosilyticus* and *Ba. eggerthii*.

Other bacterial genera besides *Bacteroides* or *Bifidobacterium* were detected at a PID of 50 to 60, especially to the three glucosidase sequences of *Bi. pseudocatenulatum* (Figure 1). At PID values of 55 to 60 to BpGluE, MAGs from *Faecalicatena gnavus* and the unclassified *Gemmiger* sp003476825 were highly abundant, with 3170 and 8329 overall occurrences, respectively (Appendix A). No *Gemmiger* bacterium was associated with the deglycosylation of flavonoids up to now, but the high PID to BpGluE and the highest overall abundance of *O*-glycosidase hits make *Gemmiger* sp003476825 (one of the most prevalent bacteria in the human gut [61]) a potentially highly relevant species for flavonoid deglycosylation. Two other not-taxonomically-described species with a high number of *O*-glycosidase hits and a PID of more than 50 were CAG-180 sp000432435 (Clostridiales, family *Acutalibacteriaceae* about 2000 occurrences) and CAG-217 sp000436335 of the same family with about 1000 occurrences. All of these show a PID of about 50 to *Bi. pseudocatenulatum* GluD. The high abundance and similarity to BpGluD indicates a potentially prominent role of these *Acutalibacteriaceae* members in deglycosylation of flavonoids. Furthermore, *Blautia wexlerae* and two taxonomically undescribed *Eubacterium* spp. with a high PID to BpGluD were abundant as well. More distantly related glucosidases (PID between 40 and 50 to the queries) include those of *Faecalicatena gnavus*, several *Clostridium* and *Ruminococcus* species and *Gemmiger* sequences more closely related to the *Bifidobacterium breve* enzyme and a second *Blautia* enzyme (Appendix A). Since only isoflavones were tested as substrates for the bifidobacterial enzymes, conclusions about the substrate specificity of the homologs detected here cannot be given. Usually, flavonones and flavonols are also deglycosylated by *Bifidobacterium* spp. [11].

To unravel the phylogenetic relationships between the observed enzymes and to see whether the novel sequences can be classified into distinct phylogenetic classes with potential new catalytic properties, a phylogenetic tree was created. The largest part of the *Bl. wexlerae* sequences form a large distinct clade that also includes sequences from *Facealicatena gnavus*, while the *Gemmiger* glucosidases forms another clade, also distinct from all clades harboring characterized flavonoid *O*-glycosidases (Appendix A). The latter forms one phylogenetic clade together with glucosidases of four *Acutalibacteraceae* species. Only four of the sequenced enzymes (BpGluA, BpGluB, BpGluD) and the *Bacteroides* sequence form larger clades, with the *Bl. wexlerae*, the CAG-180 and the *F. prausnitzii* clades being much larger, suggesting that the *O*-deglycosylating potential of the human gut microbiota is largely uncharacterized and awaits biochemical investigation.

A second class of *O*-glycosidases was observed in *Catenibacillus scindens*, where a cluster of two genes, *dfgCD,* was characterized to encode an enzyme that 7-*O*-deglycosylated isoflavones and flavones [17]. DfgCD homologs are found scarcely in human gut MAGs, mainly in *Hungatella hathewayi*, *Faecalicatena* species, and *Eisenbergiella* sp900066775, all with approximately 100 occurrences (Appendix A).

### 2.4. Derhamnosylation of Flavonoids

The BLAST search with queries from six rhamnosidases of *Lactobacillus*, *Bifidobacterium* and *Bacteroides* species resulted mainly in hits with MAGs classified as *Bifidobacterium* and *Bacteroides*. The low number of hits to *Lactobacillus* MAGs (only *L. plantarum* was detected with a PID > 65 and Freq > 20) was expected, given the low abundance of *Lactobacillus* species in the gut MAGs metadata (Table 2). One of the most abundant species was *Bacteroides dorei*, not yet described as a flavonoid rhamnoglycoside-hydrolyzing species (Figure 2), whereas the species described to de-rhamnosylate, e.g., rutin, such as *Bi. pseudocatenulatum*, *Bi. dentium*, *Bi. breve*, and *Ba. thetaiotaomicron*, were of comparably lower numbers. The highest number of MAGs in this analysis was observed when using the metagenome-derived rhamnosidase 3 sequence as a query. These MAGs could be assigned to *Gemmiger* sp003476825, which also contains a putative flavonoid *O*-glucosidase (see above, no sequence similarities to the rhamnosidase described here). With a lower abundance (about 50 appearances), metagenome-derived rhamnosidase 2 orthologs were detected in MAGs of the genus CAG-170 (family *Oscillospiraceae*). Orthologs (PID of about 98) of the metagenome-derived rhamnosidase 1 were detected in only five MAGs from the family *Monoglobaceae*, which could not yet be assigned to a genus and differ from the only described member of this genus, the pectin-degrading intestinal *Monoglobus pectinilyticus* [65,66], which does not encode the corresponding rhamnosidase. Below a 65 PID cutoff, mainly species from the genera *Parabacteroides* and *Bacteroides* and the species *Alistipes onderdonkii* (approximately 45 to 50 PID to MGR2) were found to be highly abundant (Appendix A). Several distinct rhamnosidase families can be distinguished phylogenetically, of which the largest cluster does not include characterized ones and contains mainly sequence hits from the *Bacteroides* and *Parabacteroides* genomes. Interestingly, most characterized rhamnosidases do not belong to phylogenetic nodes that mainly represent the corresponding species: *B. dentium* is assigned to a cluster containing mostly *B. pseudocatenulatum* genomes, *L. acidophilus* belongs to a large node containing mostly *Fusicatenibacter saccharivorans*, the node including the characterized *Bifidobacterium breve* rhamnosidase is dominated by *Parabacteroides merdae*, while the LpR2 mainly produced hits from *Bacteroides dorei* and *Parabacteroides species* (Appendix A). The *B. thetaiotaomicron* rhamnosidase and LpR1 cluster together in a large node dominated by *Bacteroides dorei*. These results show that the characterized isolate species often do not represent the species that are mainly responsible for the de-rhamnosylating activity in the human gut.

### 2.5. C-Deglycosylation of Flavonoids

The current gene clusters reported to play a role in *C*-deglycosylation are depicted in Figure 3. Of the *Eubacterium cellulosolvens dfgABCDE* gene cluster (Figure 3A, [17]), all sequences were considered as queries in this study because the question as to which gene products are involved in catalysis has not been completely resolved. Of the *dgp* gene cluster of strain PUE (Figure 3B), we used only the first three genes (*dgpABC*) as queries, since only these genes were shown to catalyze reactions in the *C*-deglycosylation pathway [25,26].

To respond to the involvement of more than one enzyme in this reaction, we filtered for genomes encoding all three catalytic subunits *dgpABC* or at least three of *dfgABCDE*. The latter resulted in only a few similar hits with a PID higher than 40 (less than 100, Appendix A), most of them in the *Faecalicatena* genus (Appendix A). Since *dfgD* was not found in the majority of MAGs containing *dfgABCE*, it is likely that DfgD is not involved in *C*-deglycosylation. The effect of a lack of *dfgD* in the *dfgABCDE* gene cluster in *C*-deglycosylation was not specifically investigated [17]. A higher number of hits to DfgA (more than 300) with a PID lower than 40 was observed in *Faecalicatena* species, but these were single genes that were not part of a *dfgABCDE* cluster (Appendix A). The *dgpABC* cluster was much more prominent in the human gut MAGs than the *dfgABCE* cluster, with most hits in *Agathobacter faecis* (formerly *Roseburia faecis*, PID of 50 to 85%) plus several dozen in *Blautia*, *Dorea*, *Enterococcus* and *Faecalicatena* MAGs (Figure 4). However, only a low percentage of *A. faecis* MAGs carried the *dgpABC* cluster, i.e., 804 of the 2819 MAGs assigned to that species (~30%), so the occurrence of *A. faecis* cannot necessarily be linked to the *C*-deglycosylation of flavonoids. The occurrence of *C*-deglycosylating genes in *Faecalicatena gnavus* seems even less specific, with only 84 MAGs of 1197 encoding DgpABC (Appendix A).

### 2.6. Daidzein-to-Equol Conversion

The three enzymes responsible for daidzein-to-equol conversion (DZR, DDR and TDR) plus the optional dihydrodaidzein racemase encoded by *Lactococcus garviae* [30] are encoded in a single cluster, which is very similar in *L. garviae*, *A. equolifaciens* and *S. isoflavoniconvertens*. Due to the high sequence similarities of the enzymes from the three species, we used only the *S. isoflavoniconvertens* sequences (plus the *L. garviae* racemase sequence) as queries. Up- and downstream genes were not considered in this study, since they were suggested to play accessory roles [29]. Hits in *Slackia*, *Lactococcus* or *Eggerthella* were scarce, which contradicts reports in the literature on a widely distributed daidzein-to-equol conversion by these species in the human gut. Four hits of *Slackia* MAGs containing the daidzein-to-equol gene cluster were assigned to the same species represented by GUT_GENOME145587. A comparison of this representative genome to the available *S. isoflavoniconvertens* DSM22006 genome (GenBank assembly number GCA_003725955.1) using ANI calculator [67] resulted in an ANI value of 95%. Therefore, the classification of these MAGs to *S. isoflavoniconvertens* is justified. Since the other 261 MAGs representing this species do not encode the daidzein-to-equol pathway enzymes, *S. isoflavoniconvertens* appears not to be a typical equol producer in the human gut. Therefore, conclusions of equol production in an individual should not be based on 16S rRNA analyses, which obviously cannot resolve the daidzein-to-equol transforming potential of *Slackia* spp. Another bacterial species described to transform daidzein to equol, *Adlercreutzia equolifaciens*, was more abundant, with 33 occurrences of the *ddr* gene, but also here, the percentage of MAGs carrying the daidzein-to-equol gene cluster was relatively low with less than ten percent (a total of 271 *A. equolifaciens* MAGs). This observation is strengthened by a report of a human gut *A. equolifaciens* isolate that lacks the cluster and is not able to transform daidzein to equol (Valezquez et al., 2020). Another species whose MAGs frequently carried daidzein-to-equol cluster genes was CAG-1427 sp000435475 (*Eggerthellaceae*, Figure 5), of which nearly 50% (a number of 35) of the MAGs (Appendix A) carry the daidzein-to-equol gene cluster. Therefore, the occurrence of these bacterial species cannot give a reliable conclusion on the main equol producers in the human gut. Specifically designed primers in a PCR-based method revealed that three of 17 arbitrarily selected individuals (of which nine were “equol-producers”) carried equol-producing *Slackia*, and another four carried *Adlercreutzia* [52], which together with the present study hints towards an at least equal or higher importance of *A. equolifaciens* compared to *S. equolifaciens* in equol production. However, the higher percentage of equol-producing individuals is in contrast to the low number of MAGs from potential equol-producing bacteria in the present study. This might be explained by a low abundance of equol-producers in most individuals, preventing an efficient assembly of MAGs from metagenomes, which of course is much less sensitive compared to sensitive HPLC-based equol detection or the highly specific PCR-based detection of the corresponding genes. Since two of the equol-producing individuals of the mentioned study did not result in a PCR product with the employed primer pairs, undiscovered equol-forming bacteria exist in the human gut. A high number of hits with a PID of 30 to 40 was identified in *Agathobacter rectalis* (previously *Eubacterium rectale*) and several, mostly uncharacterized, *Collinsella* species (Figure 5). The occurrence of this latter genus in human gut microbiomes correlated with the ability to transform daidzein to equol in several studies [68,69,70]. However, since the PID to the characterized daidzein-transforming enzymes is rather low, with values of only about 30 to 35, and the corresponding genes are not clustered, but spread out in the genome, this assumption warrants further research. A relatively high number of genomes showing only hits to the daidzein reductase sequence might be responsible for the large number of phenotypes transforming daidzein only to *O*-Desmethylangolensin. In particular, bacteria assigned to *Erysipelatoclostridium* and *Clostridium* M genera carry putative *dzr* genes (Appendix A, Appendix A).

### 2.7. Flavone/Flavonol Reduction, Flavanone/Flavanonol Ring Cleavage, Chalcone Isomerization and Phloretin Hydrolysis

Since these four reactions, the reduction of flavons and flavonols to flavanones and flavanonols, respectively (catalyzed by Flr), the C-ring cleavage of these products (via Fcr), isomerization of chalcones and auronols (via CHI), and the hydrolysis of the dihydrochalcone phloretin (via Phy) can be viewed as steps in the same pathway of flavone/flavonol degradation (e.g., of apigenin via naringenin, Figure 6), the results are compiled in one subchapter.

The very recently characterized Flrs from *Flavonifractor plautii* and *Clostridium ljungdahlii* [50] resulted in a large number of hits in *F. plautii* and *Clostridoides difficile* MAGs (the latter interestingly with a higher PID to the sequence of *F. plautii* than to the sequence of *C. ljungdahlii*), and, with a lower PID, in MAGs of the *Olsenella, Eubacterium* and *Fusobacterium* species (Figure 7). Most of the *F. plautii* MAGs (and also the strain type DSM 6740) harbor two copies of the *flr* gene, with one of them more closely related (PID of ~60) to that of *Clostridium ljungdahlii* than to the “main” Flr of *F. plautii*. This second putative Flr of *F. plautii* did not show any apigenin-reducing activity [50], but might reduce other flavonoid classes, which were not tested in the corresponding study.

The BLAST analysis with the characterized flavanone- and flavanonol-cleaving reductase (Fcr) from *Eubacterium ramulus* resulted in hits identified in *F. plautii* MAGs (523 overall occurrences, Appendix A), while *E. ramulus* MAGs revealed a lower number of 188 occurrences. Similar to the situation with Flr, several *E. ramulus* genomes contained two *fcr* genes, one of which had a higher PID to the *F. plautii* gene, which might reflect a different substrate spectrum of the encoded Fcrs. Besides *F. plautii* and *E. ramulus*, MAGs assigned to *Anaerostipes hadrus* frequently (138 MAGs) encoded an Fcr-like enzyme with approximately 85% identity to Fcr from *E. ramulus*. Besides these three main species, a number of MAGs from *Clostridium boltae* encoded a putative Fcr with a PID of approximately 42 to the Fcr from *F. plautii*. Most hits to Fcr (several hundred up to nearly 1000) showed around 35 PID (Figure 8) and frequently occurred, in particular, in *Faecalicatena* spp. Whether these putative enzymes have a similar function and whether *Faecalicatena* and other species contribute to flavonoid degradation warrants further investigation.

Genes similar to that encoding CHI (Appendix A) were highly abundant in MAGS assigned to *Flavonifractor plautii* (593 overall occurrences) and *Clostridoides difficile* (375) and, to a lesser extent, were present in *Eubacterium ramulus* (137), in which it was initially described. Below a PID of 40, only MAGs from *Fusobacterium mortiferum* were detected in a higher number (ca. 100).

The BLAST search using the phloretin hydrolase (Phy) sequence of *E. ramulus* resulted in several hits with a PID of more than 90, assigned to 159 *E. ramulus* MAGs (Figure 9). The most abundant Phy homologs with a PID of 46 (579 occurrences, Appendix A) were found in MAGs assigned to *Flavonifractor plautii*, which was described to hydrolyze phloretin [71]. Genes of two species never reported to transform flavonoids, *Dialister succinatiphilus* (322 occurrences overall) and *Anaerostipes hadrus* (119 occurrences), were abundant with a PID of approximately 60 and 75 to the Phy query. Opposed to the Phy of *E. ramulus*, the BLAST search with the Phy of *M. abscessis* resulted in only few hits below 40 PID in a *Dakarella* species.

The co-occurrence of hits to all enzyme sequences (Flr, Fcr, CHI and Phy) was observed in most of the *F. plautii* and *E. ramulus* MAGs, emphasizing the co-functionality of these enzymes in a single pathway of flavone degradation. *F. plautii* was by far the most abundant species in this analysis, pointing toward its major role in flavone degradation in the human gut. Of the 610 MAGs assigned to *F. plautii*, most (509 to 610) contained at least one of the genes. In contrast, *A. hadrus*, containing only Fcr and Phy, showed only a minor amount of MAGs carrying flavonoid-degrading enzymes (less than 5% of the overall MAGs).

### 2.8. O-Demethylation of Flavonoids

Similar to *C*-deglycosylation, *O*-demethylation requires several proteins (MT1 and MT2, CP and AE) acting together in a complex reaction pathway. Since no specific flavonoid *O*-demethylation system has been characterized, the four enzyme sequences of the *O*-demethylase system from *Eubacterium limosum,* which acts on the polyphenolic lignan secoisolariciresinol [32], were used as queries. Above 70 PID, only a few dozen hits from the *Eubacterium* genus, notably from the species *E. callenderi* and to a lesser extent *E. limosum* (characterized for flavonoid *O*-demethylation), were observed (Appendix A). The highest number of similar *O*-demethylase sequences was detected in MAGs of *Intestinibacter bartletti* (PIDs of 60 for MT1, ca. 40 to 50 for MT2 and CP; ca. 33 for AE). The high PID of MT1, which defines the substrate specificity of *O*-demethylation [34], suggests that *I. bartlettii* (never tested for *O*-demethylation of flavonoids) has a similar substrate spectrum for *O*-demethylation as *E. limosum*. The MTI gene is part of a complete *O*-demethylase gene cluster in *I. bartlettii* (Appendix A), including two CP genes and *cobW*, which encodes a cobalt-binding protein. The high number of MAGs with hits to the CP of the *O*-demethylase operon sequences (Appendix A) can be explained by the high abundance of proteins homologous to CP due to the common B_12_-binding and Fe-S cluster domains. In fact, only a low number of MAGs were found to contain the complete operon with a PID of above 40, with *I. bartlettii* MAGs being the most numerous (69 occurrences, Appendix A).

## 3. Conclusions

The in silico analysis of potential flavonoid-modifying enzymes of human gut bacteria presented here is a first step to uncover the complete picture of flavonoid transformation in the human intestine. The findings constitute a basis for targeted testing of the detected potential novel enzymes and their bacterial sources. Figure 10 summarizes some representative flavonoid-transformations, the prevalence of MAGs encoding the corresponding flavonoid-transforming enzymes and the main bacterial species involved in these reactions. In addition to the already characterized gut bacteria involved in flavonoid transformation, potential species and genera so far not associated with the modification of these polyphenols in the human gut were identified. Of these novel flavonoid-modifying candidates, several species are as-yet not isolated. These include, for example, *Gemmiger* sp003476825, which is potentially involved in *O*-deglycosylation and derhamnosylation, or the CAG-1427 genus, which could be involved in *O*-deglycosylation. Heterologous expression of the potentially involved enzymes could reveal their flavonoid deglycosylation potential. Some other bacteria were already known to play a crucial metabolic role in the human gut (e.g., production of short-chain fatty acids or polysaccharide degradation), but their ability to convert flavonoids may have been overlooked. An example is *Anaerostipes hadrus*, so far mainly known for butyrate production [72]. An (albeit minor) portion of MAGs of this highly abundant human gut bacterium was observed to encode several putative flavonoid-transforming enzymes including a Phy and an Fcr. A major flavonoid-degrading bacterium seems to be *Flavonifractor plautii*. Most of its MAGs contain the Flr, Fcr, Phy and CHI genes, and it is relatively abundant. Thus, *F. plautii* might serve as a good indicator of the flavonoid degradation potential in 16S rRNA gene-based gut microbiota analyses. On the other hand, our quantitative results on flavonoid-modifying bacteria imply that several key flavonoid modification processes in the human gut, which might have been so far assigned to bacteria, are neither representative nor the main flavonoid-transforming bacteria in this ecosystem. For example, equol formation from daidzein appears to mainly be catalyzed in the human intestine by *Eggerthella*, and it may be catalyzed by *Collinsella* species more frequently than by the thoroughly studied *Slackia isoflavoniconvertens* and *Adlercreutzia equolifaciens*.

The observation of the wide array of uncharacterized, putative flavonoid-transforming enzymes in human gut bacteria presented here might be a valuable source for identification of novel enzymes of biotechnological interest, especially for the production of unusual or difficult-to-produce flavonoids. Several bacterial species seem to be specialized in metabolizing flavonoids and possibly other polyphenols as well. For example, nearly all *Flavonifractor plautii* MAGs harbor the three enzymes Phy, Fcr and CHI, and therefore seem to be the main flavanone and flavanonol-degrading bacteria in the human gut. While further research on the effect of this degradation on human health is necessary, it might be one of the reasons why the presence of *F. plautii* was associated with the development of colon cancer in an Indian population [73]. Finally, the outcome of the in silico analysis presented here could contribute to the clarification of the impact of gut bacteria on flavonoid-mediated health effects in humans.

## 4. Methods

### 4.1. Retrieval of Query Enzyme Sequences, MAGs Data and Set-Up of UHGP BLAST Database

Besides extracting the most recent review on flavonoid modification [11], a literature search on the databases from Web of Science and Pubmed was performed. In addition, a Google scholar search was employed. The keywords used were “flavonoid(s)” AND “modification” OR “transformation” OR “deglycosylation” OR “demethylation” OR “cleavage”. Furthermore, screening in the citation network of the retrieved publications was performed and manually reviewed before Genbank accession numbers and the corresponding amino acid sequences were retrieved from the NCBI Genbank database (Table 1).

A BLAST-searchable database including all non-redundant amino acid sequences from the UHGP catalog was set up as follows: The UHGP-100 version 1.0 non-redundant protein fasta file (uhgp-100.faa) from 286,997 MAGs assembled from gut metagenomes [61] was downloaded as a packed uhgp-100.tar.gz file from the MGnify ftp server (http://ftp.ebi.ac.uk/pub/databases/metagenomics/mgnify_genomes/human-gut/v1.0/, accessed on 1 July 2020) and subsequently extracted. With BLAST+ (stand-alone, version 2.9.0 [74]) installed on Linux Ubuntu 18.04 and the makeblastdb command, a BLAST database was created from the uhgp-100 file. For quantification purposes, the uhgp-100.tsv table, including all of the redundant amino acid sequence IDs (all assigned to a non-redundant “representative” sequence ID corresponding to the identical amino acid sequence in other MAGs), was extracted from the same packed file (uhgp-100.tar.gz). The UHGP metadata, which included MAGs taxonomy assigned via the genome taxonomy database (GTDB, release 89), genome quality values, study and sample information and others, were downloaded from the same mgnify ftp-address (genomes-all_metadata.tsv file). Please note that there might be differences in the taxonomy, since the current release of the GTDB is no. 202 and the metadata file was not updated.

### 4.2. BLAST Searches, Data Mangling and Quantification

The queries were searched against the UHGP-100 BLAST database using the BLASTP search function and an e-value cutoff of 1e-25 for pathways including short sequences (150 to 350 amino acids; daidzein-to-equol, CHI, O-demethylation, Phy, Flr, DfgCD) or 1e-60 for longer sequences (>350 amino acids; Fcr, *O*-glycosidases, rhamnosidases) and 1e-20 for the *C*-deglycosylation enzymes, which contained two sequences below 150 amino acids. The maximal number of target hits was set to 100,000 via the max_target_seqs argument. Blast table output format 6 was chosen with the specified columns (the Linux BASH commands are given in the Appendix A). The output files were concatenated and filtered for a PID of at least 30 and a query coverage value of at least 75%. To avoid redundant hits to more than one query, only the hits with the best bitscore were kept. To retrieve the redundant amino acid sequence IDs for quantification of the hits, the grep command was used on the uhgp-100.tsv file with a list of the non-redundant sequence IDs as input. The resulting table was merged with the metadata and subsequently filtered for MAGs only (isolate genomes also included in the uhgp catalog were discarded) and the occurrences of the redundant amino acid sequences were counted; the values accounting for the frequency in a given non-redundant sequence group were placed into a new column (designated Freq). For further filtering and data visualization, R (v 3.6.3) was used via RStudio (v 1.3), employing the package ggplot2, v.3.3.0 [75].

### 4.3. MAGs Gene Cluster Analysis

To compare and analyze gene clusters among MAGs or isolate genomes, the corresponding genomes were downloaded from the NCBI database or the UHGP MAGs folder and uploaded to the rapid annotation subsystem technology (RAST) server (with their original annotation preserved if possible). The sequence comparison tool was used to compare up to ten genomes.

### 4.4. Construction of Phylogenetic Trees

Phylogenetic trees of the rhamnosidase and O-glycosidase sequences were constructed after clustering of the sequences with CD-hit (v 4.8.1 [76]) using standard settings (90% PID clustering threshold) and sequence alignment with ClustalOmega (v 1.2.4, [77]) installed on Linux Ubuntu. The alignment was refined with Clipkit (v 1.1.2, [78]) before the tree was generated with Fasttree (v 2.1.10, standard settings [79]). Tree visualization was performed via iTol (v 6, [80]) and manually refined in Inkscape for Windows (v 1.01). The size of the node triangles was drawn to scale using the number of CD-hit clustered sequences as well as the redundant sequence IDs of the uhgp-100.tsv file.

## Figures and Tables

**Figure 1 nutrients-13-02688-f001:**
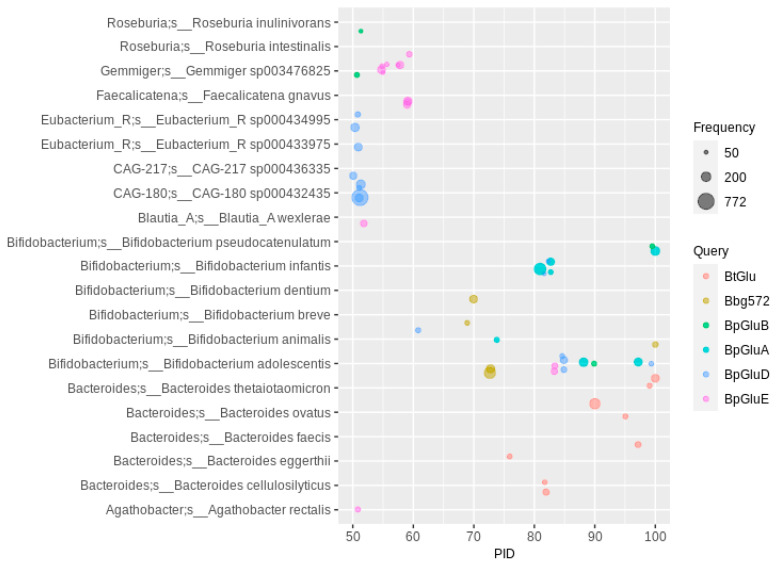
Flavonoid *O*-glycosidase homologs in human gut MAGs. The PID (percent amino acid sequence identity) threshold to the queries (see color code) was set to 50 (For abbreviations and details, see Table 1). Hits were filtered for at least 50 occurrences, so that each bubble represents a number of redundant sequences ranging from 50 to 772.

**Figure 2 nutrients-13-02688-f002:**
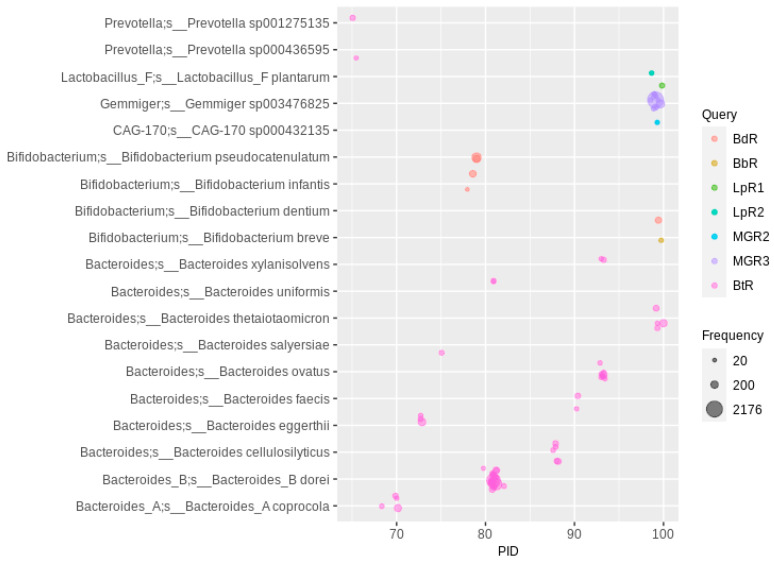
Rhamnosidase homologs in human gut MAGs. A PID threshold of 65 was chosen for the queries shown in the color code (for abbreviations and details, see Table 1; hits with a lower PID are shown in Appendix A). Hits were filtered for at least 20 occurrences, so that each bubble represents a number of redundant sequences ranging from 20 to 2176.

**Figure 3 nutrients-13-02688-f003:**
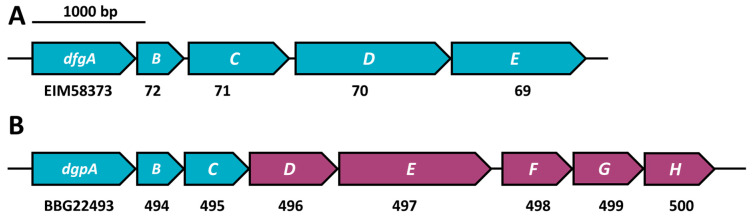
Two characterized *C*-deglycosylation gene clusters. (**A**) *Eubacterium cellulosolvens* with five genes involved in *C*-deglycosylation [17] and (**B**) strain PUE [25,26], with *dgpA* encoding an oxidoreductase and *dgpBC* as oxo-puerarin-deglycosylating enzymes, of which all three are involved in *C*-deglycosylation. Genes encoding deglycosylating enzymes are shown in turquoise, accessory genes in violet. The full GenBank accession number is given for the first gene and only variable digits are given for the downstream genes.

**Figure 4 nutrients-13-02688-f004:**
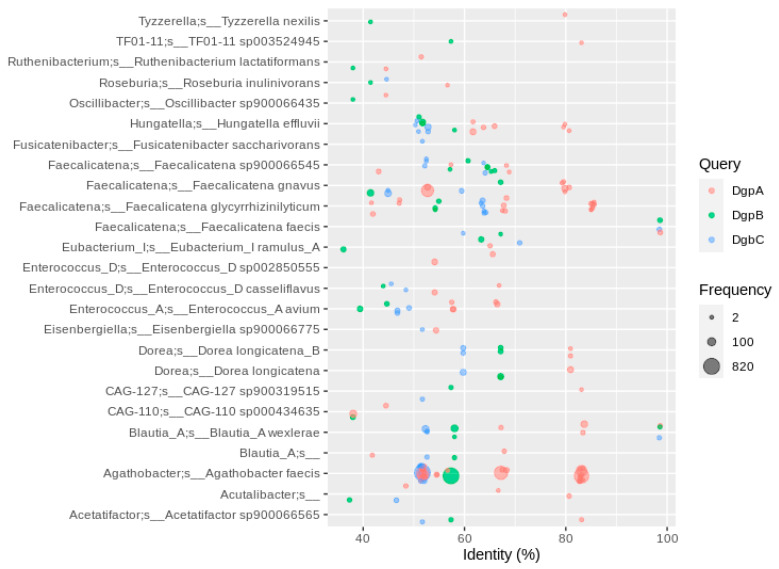
Homologs to enzymes involved in flavonoid *C*-degylcosylation using sequences of strain PUE. Hits were filtered for the co-occurrence of all three genes required for *C*-deglycosylation (*dgpABC*) in the same MAG. Hits were filtered for at least 2 occurrences, so that each bubble represents a number of redundant sequences ranging from 2 to 820.

**Figure 5 nutrients-13-02688-f005:**
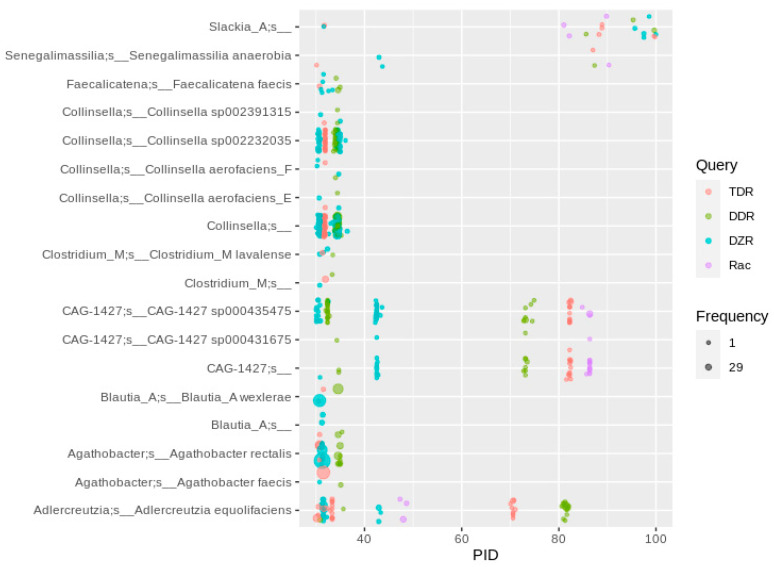
Homologs to enzymes involved in daidzein-to-equol transformation in human gut MAGs. Hits were filtered for the co-occurrence of all three genes required for daidzein-to-equol conversion (*dzr*, *ddr*, *tdr*) in a single MAG. For a plot showing the hits to individually occurring genes (mainly *dzr*), see Appendix A.

**Figure 6 nutrients-13-02688-f006:**
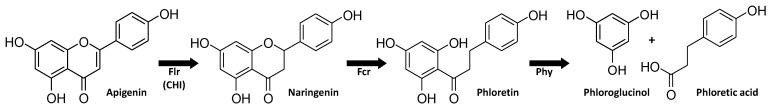
Reductive degradation of flavonoids depictured with apigenin as a flavone example.

**Figure 7 nutrients-13-02688-f007:**
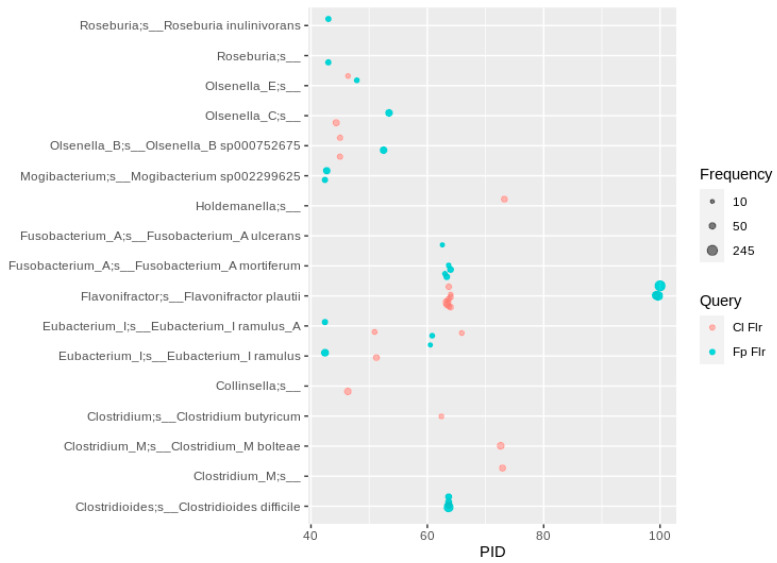
Flr sequence hits of human gut MAGs. A PID threshold of 40 was chosen to the queries shown in the color code (for abbreviations and details see Table 1). Hits were filtered for at least 10 occurrences, so that each bubble represents a number of redundant sequences ranging from 10 to 245.

**Figure 8 nutrients-13-02688-f008:**
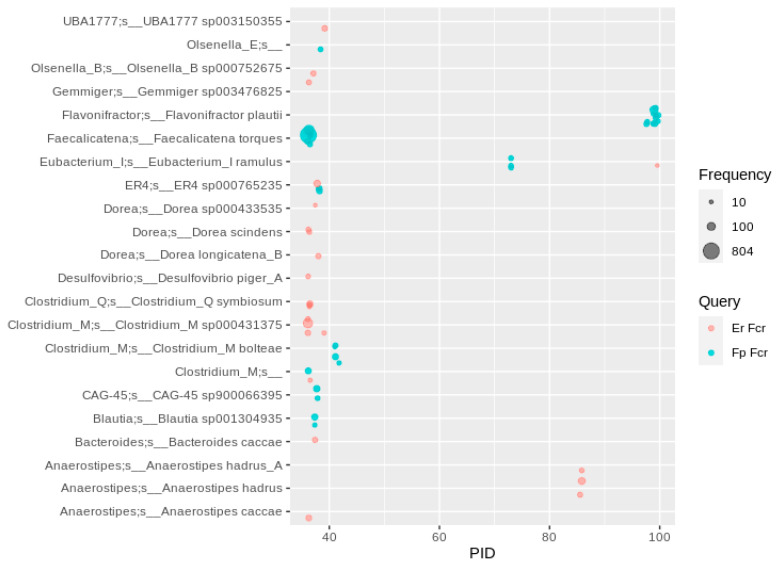
Fcr-like enzymes in human gut MAGs. Hits were filtered for at least 10 occurrences, so that each bubble represents a number of redundant sequences ranging from 10 to 804.

**Figure 9 nutrients-13-02688-f009:**
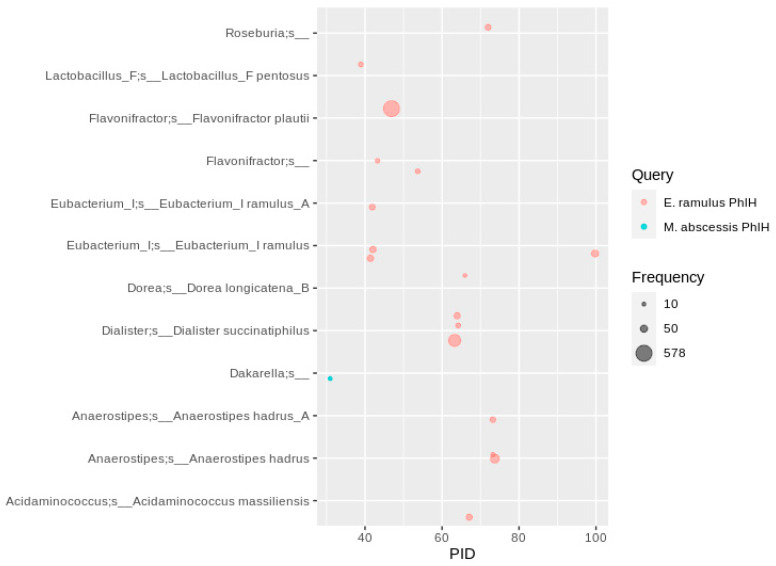
Distribution of Phy homologs in human gut MAGs. Depicted are MAGs with at least five identical amino acid sequences. Hits were filtered for at least ten occurrences, so that each bubble represents a number of redundant sequences ranging from 10 to 578.

**Figure 10 nutrients-13-02688-f010:**
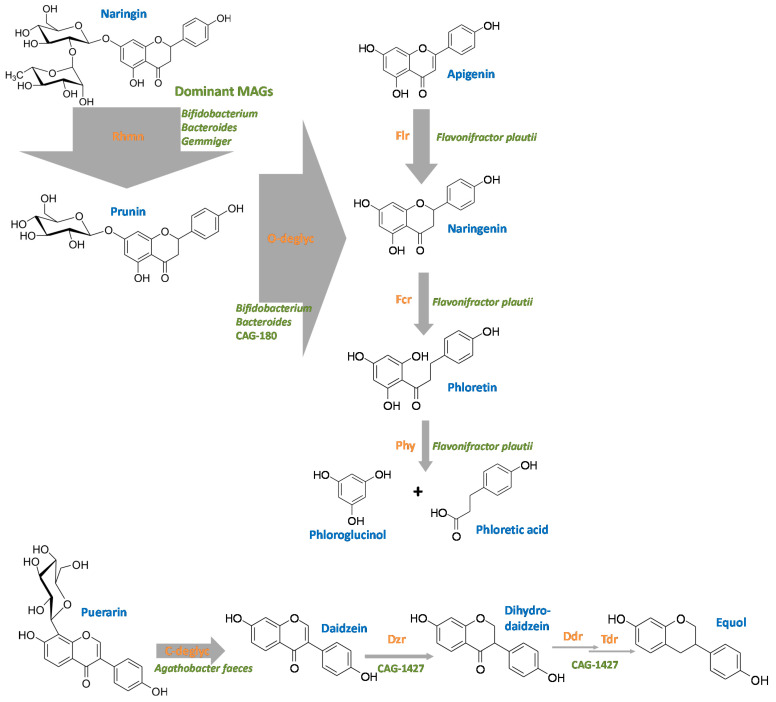
Overview of the flavonoid conversion pathways in the human gut based on the MAGs study. Thickness of arrows reflects the abundance of observed hits to the sequences of the characterized flavonoid-transforming enzymes. Color code: blue, flavonoid trivial names; orange, flavonoid-converting enzymes; green, most abundant bacterial species carrying out the corresponding reaction in this analysis. For abbreviations, see text.

**Table 1 nutrients-13-02688-t001:** Characterized flavonoid-modifying enzymes of human gut bacteria.

Accession No.	Enzyme	Reaction	Source Bacterium	Reference
KEF29323.1	Beta-glucosidase BpBluA	7-*O*-deglycosylation of isoflavones	*Bifidobacterium pseudocatenulatum* IPLA36007	[15]
KEF27912.1	Beta-glucosidase BpGluB	7-*O*-deglycosylation of isoflavones	*Bifidobacterium pseudocatenulatum* IPLA36007	[15]
KEF28010.1	Beta-glucosidase BpGluD	7-*O*-deglycosylation of isoflavones	*Bifidobacterium pseudocatenulatum* IPLA36007	[15]
KEF28001.1	Beta-glucosidase BpGluE	7-*O*-deglycosylation of isoflavones	*Bifidobacterium pseudocatenulatum* IPLA36007	[15]
AFS33105.1	Beta-glucosidase Bbg572	7-*O*-deglycosylation of isoflavones, 3-/4-/7-*O*-deglycosylation of quercetin glycosides	*Bifidobacterium animalis* ssp. *lactis* SH5	[16]
AAO76887.1	Beta-glucosidase BtGlu	7-*O*-deglycosylation of isoflavones	*Bacteroides thetaiotaomicron* VPI-5482	[17]
AGS77942.1	Alpha-L-Rhamnosidase BdR	Derhamnosylation of 1→2 and 1→6 *O*-glycosidic bond of rutinosylated flavonoids	*Bifidobacterium dentium* K13	[19]
AHJ22585.1	Alpha-L-Rhamnosidase BbR	Derhamnosylation of 1→6 *O*-glycosidic bond of rutin	*Bifidobacterium breve* 689b	[20]
WP_011107561	Alpha-L-Rhamnosidase BtR	Derhamnosylation of 1→2 *O*-glycosidic bonds of rutinosylated flavonoids	*Bacteroides thetaiotaomicron* VPI-5482	[21]
QBM20340.1	Alpha-L-Rhamnosidase HFM-RhaA (MGR1)	Derhamnosylation of 1→2 and 1→6 *O*-glycosidic bond of rutinosylated flavonoids	Human gut metagenome	[24]
QBM20341.1	Alpha-L-Rhamnosidase HFM-RhaB (MGR2)	Derhamnosylation of 1→2 *O*-glycosidic bond of rutinosylated flavonoids	Human gut metagenome	[24]
QBM20342.1	Alpha-L-Rhamnosidase HFM-RhaC (MGR3)	Derhamnosylation of 1→2 and 1→6 *O*-glycosidic bond of rutinosylated flavonoids	Human gut metagenome	[24]
CCC80440.1	Alpha-L-Rhamnosidase LpR1	Derhamnosylation of 1→6 *O*-glycosidic bond of rutinosylated flavonoids	*Lactobacillus plantarum* WCFS1 *	[22]
CCC80442.1	Alpha-L-Rhamnosidase LpR2	Derhamnosylation of 1→6 *O*-glycosidic bond of rutinosylated flavonoids	*Lactobacillus plantarum* WCFS1 *	[22]
AAV43293.1	Alpha-L-Rhamnosidase LaR	Derhamnosylation of 1→2 and 1→6 *O*-glycosidic bond of rutinosylated flavonoids	*Lactobacillus acidophilus* NCFM	[22]
EUA80835.1	Phloretin hydrolase	Phloretin hydrolysis	*Mycobacteroides abscessus* ssp. *bolletii* 103	[28]
AAQ12341.1	Phloretin hydrolase	Phloretin hydrolysis	*Eubacterium ramulus* DSM 16296	[29]
AKC35075.1 AKC35076.1	DfgCD	7-*O*-deglycosylation of isoflavones and flavones	*Catenibacillus scindens* DSM 106146	[25]
EIM58373.1 EIM58372.1 EIM58371.1 EIM58370.1 EIM58369.1	DfgABCDE	*C*-deglycosylation of flavones	*Eubacterium cellulosolvens* ATCC 43171	[25]
BBG22493.1 BBG22494.1 BBG22495.1	DgpABC	*C*-deglycosylation of isoflavones	*Dorea* sp. strain PUE	[26]
ANU40626.1	Flavone/flavonol reductase (Flr)	C-ring double bond reduction	*Flavonifractor plautii* DSM 6740	[50]
ADK16070.1	Flavone/flavonol reductase (Flr)	C-ring double bond reduction	*Clostridium ljungdahlii* DSM 13528	[50]
AGS82961.1	Flavanone/flavanonol-cleaving reductase (Fcr)	Reductive ring cleavage of flavanones and flavanonols	*Eubacterium ramulus* DSM 16296	[45]
WP_154024723.1	Flavanone/flavanonol-cleaving reductase (Fcr)	Reductive ring cleavage of flavonones and flavanonols	*Flavonifractor plautii* DSM 6740	[45]
AIS36173.1	Chalcone isomerase (CHI)	Isomerization of chalcones and flavanonols	*Eubacterium ramulus* DSM 16296	[46,47,49]
EHM54434.1	Chalcone isomerase (CHI)	Isomerization of chalcones	*Flavonifractor plautii* ATCC 29863 (formerly *Clostridium orbiscindens)*	[46,47,49]
AFV15450.1	Tetrahydrodaidzein reductase (TDR)	Reduction of tetrahydrodaidzein	*Slackia isoflavoniconvertens* DSM 22006	[30]
AFV15451.1	Dihydrodaidzein reductase (DDR)	Reduction of dihydrodaidzein	*Slackia isoflavoniconvertens* DSM 22006	[30]
AFV15453.1	Daidzein reductase (DZR)	Reduction of daidzein	*Slackia isoflavoniconvertens* DSM 22006	[30]
BAL46928.1	Tetrahydrodaidzein reductase (TDR)	Reduction of tetrahydrodaidzein	*Slackia* sp. NATTS	[32]
BAL46929.1	Dihydrodaidzein reductase	Reduction of dihidydrodaidzein	*Slackia* sp. NATTS	[32]
BAL46930.1	Daidzein reductase	Reduction of daidzein	*Slackia* sp. NATTS	[32]
BAM25050.1	Dihydrodaidzein Racemase	Racemization of dihydrodaidzein	*Lactococcus garviae* 20-92	[31]
BAJ72744.1	Tetrahydrodaidzein reductase	Reduction of tetrahydrodaidzein	*Lactococcus garviae* 20-92	[51]
BAJ72745.1	Dihydrodaidzein reductase	Reduction of didydrodaidzein	*Lactococcus garviae* 20-92	[51]
BAJ22678.1	Daidzein reductase	Reduction of daidzein	*Lactococcus garviae* 20-92	[39]
WP_013979960.1	Tetrahydrodaidzein reductase	Reduction of tetrahydrodaidzein	*Eggerthella* sp. YY7918	[38]
WP_013979959.1	Dihydrodaidzein reductase	Reduction of dihidydrodaidzein	*Eggerthella* sp. YY7918	[38]
WP_013979957.1	Daidzein reductase	Reduction of daidzein	*Eggerthella* sp. YY7918	[38]
ANI69959.1	O demethylase (ODem) activating enzyme (AE)	Activation of CP	*Eubacterium limosum* ZL-II	[33]
ANI69960.1	ODem Methyltransferase (MT) 1	*O*-Demethylation of secoisolariciresinol	*Eubacterium limosum* ZL-II	[33]
ANI69961.1	ODem Corrinoid protein (CP)	Methyl transfer	*Eubacterium limosum* ZL-II	[33]
ANI69962.1	ODem MT2	Methyl transfer to CP	*Eubacterium limosum* ZL-II	[33]

* An identical rhamnosidase was identified in *Lactobacillus plantarum* NCC245 (Avila et al., 2009).

**Table 2 nutrients-13-02688-t002:** Flavonoid-modifying gut bacterial species and their prevalence (percent of MAGs from all samples involved) as MAGs derived from the taxonomic classification in the metadata of the UHGG [61].

Species	Flavonoid Class	Prevalence
*O*-Deglycosylation		
*Bacteroides ovatus*	Flavonols	3.1
*Bacteroides uniformis*	Flavonols, flavanones	18.5
*Bacteroides thetaiotaomicron* ^1^	Isoflavones	1.9
*Bifidobacterium adolescentis*	Flavanones, isoflavones	7.4
*Bifidobacterium angulatum*	Isoflavones	0.3
*Bifidobacterium animalis*	Anthocyanidins, isoflavones	0.3
*Bifidobacterium bifidum*	Flavanones, isoflavones	3.3
*Bifidobacterium breve*	Flavonols, isoflavones, flavanones	1.5
*Bifidobacterium catenulatum*	Flavonols, isoflavones, flavanones	0.8
*Bifidobacterium dentium*	Flavonols, flavanones	0.3
*Bifidobacterium infantis*	Flavonols, isoflavones, flavanones	8.4
*Bifidobacterium longum*	Isoflavones	0
*Bifidobacterium pseudocatenulatum*	Flavonols, flavanones	3.5
*Bifidobacterium pseudolongum*	Isoflavones	<0.1
*Blautia producta* (MRG-PMF1, 99%)	Flavonols, flavones, flavanones, isoflavones	<0.1
*Catenibacillus scindens*	Flavones, isoflavones	n.d. (<0.1)
*Enterobacter cloacae*	Flavanones	0.1
*Enterococcus avium*	Flavonols	0.1
*Enterococcus casseliflavus*	Flavonols	<0.1
*Enterococcus faecalis*	Flavanones	2.9
*Eubacterium cellulosolvens*	Flavones, isoflavones	0
*Eubacterium ramulus*	Flavonols, flavones, dihydrochalcones, isoflavones	0.5
*Escherichia* sp. HGH21 (99% to *E. coli* (MIDI *)	Isoflavones	(28)
*Escherichia* sp. 4 (*E. fergusonii*)	Flavones	<0.1
*Parabacteroides distasonis*	Flavonols, flavanones	14.5
*Lactobacillus acidophilus*	Flavanones	<0.1
*Lactobacillus buchneri*	Flavanones	<0.1
*Lactobacillus casei*	Flavanones, anthocyanidins	0
*Lactobacillus leichmanii*	Flavanones	n.d.
*Lactobacillus plantarum*	Flavanones, anthocyanidins	0.2
*Lactococcus lactis*	Flavonols, isoflavones, flavanones	0.2
*Lactococcus paracasei* [62]	Isoflavones, flavonols	0.2
***C*** **-Deglycosylation**		
*Eubacterium cellulosolvens*	Flavones, isoflavones	0
Strain PUE (1346 nt), *Dorea longicatena* (98%)	Isoflavones	n.d. (2.9)
*Catenibacillus scindens*	Flavones, isoflavones	n.d.
*Enterococcus casseliflavus* (sp. 45, 99%)	Flavones	<0.1
*Enterococcus faecium* (MRG-IFC-2, 99%)	Isoflavones	1.8
*Lactococcus lactis* (MRG-IFC-1, 99%)	Isoflavones	0.2
**C-Ring cleavage**		
*Flavonifractor plautii* (formerly *C. orbiscindens*)	Flavonols/flavanonols, flavones/flavanones	2.1
*Catenibacillus scindens*	Flavones	n.d.
*Eubacterium ramulus*	Flavonols/flavanonols, flavones/flavanones, isoflavones	0.5 ^1^
*Clostridium butyricum*	Flavanones	0.3
*Lactobacillus plantarum*	Flavan-3-ols	0.2
Strain SY8519 (*Eubacterium*_I sp000270305)	Isoflavones	n.d.
*Adlercreutzia equolifaciens*	Flavan-3-ols	0.8
*Eggerthella lenta*	Flavanonols (SDG-2), flavan-3-ols	0.9
*Enterococcus faecium* [63]	Isoflavones	1.8
**Reduction**		
*Bifidobacterium animalis*	Isoflavones	0.3
*Bifidobacterium longum*	Isoflavones	0
*Bifidobacterium pseudolongum*	Isoflavones	<0.1
*Adlercreutzia equolifaciens*	Isoflavones	0.8
*Lactococcus garvieae*	Isoflavones	<0.1
*Slackia* sp. NATTS	Isoflavones	n.d.
*Slackia isoflavoniconvertens*	Isoflavones	n.d.
*Lactobacillus rhamnosus* [64]	Isoflavones	0.4
*Enterococcus faecalis* [64]	Isoflavones	2.9
*Enterococcus faecium* [63]	Isoflavones	1.8
**Dehydroxylation**		
*Eschericha* sp. 4 (*E. fergusonii*)	Flavones	<0.1
*Adlercreutzia equolifaciens*	Flavan-3-ols	0.8
*Eggerthella lenta* (SDG-2, 99%)	Flavan-3-ols	0.9
***O*-** **Demethylation**		
*Blautia producta* (MRG-PMF1, 99%)	Flavonols, flavones, flavanones, isoflavones	<0.1
*Eubacterium limosum* (strains DSM 20543T and LMG P23546)	Isoflavones, flavanones	<0.1

Sequence identities are taken from [11] or the reference given in the species column. When the bacterium was classified into the genus, but not into a species, we performed a 16S rRNA gene analysis and list the original species/strain designation and the 16S rRNA sequence identity to the identified strain in brackets behind the species name. *: as per MIDI technique (Hur Lay et al., 2000). ^1^: This number is the combined value from two phylogenetic groups (possibly subspecies) given in the GTDB.

## Data Availability

Data are supplied as Appendix A.

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
