# Peer review of "Flavonoid-Modifying Capabilities of the Human Gut Microbiome—An In Silico Study"

_nutrients, 2021, doi:10.3390/nu13082688_

Round 1
Reviewer 1 Report
The article by Goris et al. presented an in silico analysis of potential flavonoid-modifying enzymes of human gut bacteria. The article is well organized and well presented with abundant facts and figures. Previous research mainly focused on flavonoid modifying bacterial enzymes. However, this article focused on identifying some of the flavonoid modifying gut microbiota.
Author Response
Thanks for your positive review report.
Reviewer 2 Report
This manuscript is well prepared and focuses on interesting topics.
Minor corrections may be considered.
1. The length of introduction has to be shorten than now.
2. The resolution of Figures 4, 5, 7, 8, and 9 has to be improved.
3. Although this deals with in silico study, "material and method" section has to be placed and the relevant contents should be described.
Author Response
Thanks for the review report and the constructive comments.
We uploaded a revised manuscript which includes the necessary revisions. The point-by-point responses are as follows:
- The length of introduction has to be shorten than now.
- We shortened the introduction by about 15 lines which is substantially shorter than before.
2. The resolution of Figures 4, 5, 7, 8, and 9 has to be improved.
- We will upload figures with a higher resolution in the proofing stage.
3. Although this deals with in silico study, "material and method" section has to be placed and the relevant contents should be described.
- We described all necessary steps in the "methods" part, lines 446 - 490.
Reviewer 3 Report
The article is well written and easy to understand. I advise the authors to re-read the text as there are some errors.
a 5-page introduction is perhaps a bit over the top. risks being verbose and making reading boring.
Figure 10 is not immediately understandable. The figures should be understandable without having to re-read the text. Improve figure 10.
Author Response
The article is well written and easy to understand. I advise the authors to re-read the text as there are some errors.
- Thanks for the kind words. We re-read the manuscript and corrected errors throughout the manuscript.
a 5-page introduction is perhaps a bit over the top. risks being verbose and making reading boring.
- This is true, we significantly shortened the introduction of the manuscript by more than 15 lines.
Figure 10 is not immediately understandable. The figures should be understandable without having to re-read the text. Improve figure 10.
- We revised figure 10 and introduced a color code for enzymes, bacterial species and flavonoid names to immediately capture the meaning.